## Research Article

post-traumatic stress disorder (PTSD); depression; peace of mind; healthcare workers; war zones; mental health support

**Corresponding author:**
Yaser Snoubar;
Email: ysnoubar@qu.edu.qa

# Post-traumatic stress disorder and depression as predictors of peace of mind among healthcare workers working with war victims

Ali Shakir Al-Fatlawi[1], Yaser Snoubar[2] , Yousif Saleh Mahdi[1] and Ibrahim Murtadha Alaarjy[3]

[1]Social Sciences Department (Psychology Program), Qatar University College of Arts and Sciences, Qatar; [2]Social Sciences Department (Social Work Program), College of Arts and Sciences, Qatar University, Qatar and [3]University of Baghdad, Iraq

## Abstract

Healthcare professionals in Iraq are exposed to war-related stressors that may undermine psychological well-being. This study examined post-traumatic stress disorder (PTSD) and depression and their association with peace of mind (PoM). In a cross-sectional survey, 174 physicians, nurses, pharmacists and allied health workers from multiple Iraqi regions completed an online questionnaire including demographics and validated Arabic measures of PTSD, depression (BDI) and PoM. Reliability was checked using Cronbach's alpha, and analyses used descriptive statistics, *t*-tests, Spearman correlations and stepwise regression. Mild-to-moderate PTSD symptoms were reported by 66.1% of participants, and 39.1% reported at least mild depressive symptoms; 54.0% showed moderate PoM. In regression models, higher PTSD and depression scores significantly predicted lower PoM, whereas years of professional experience predicted higher PoM. These findings indicate a substantial burden of trauma- and depression-related symptoms among Iraqi healthcare workers and suggest that workplace-focused mental health supports and organizational policies are needed to protect well-being in conflict-affected settings.

## Impact statement

Healthcare professionals in Iraq are under a combined strain of conflict, demanding clinical responsibilities and instability. The following study highlights how these conditions have a measurable impact on healthcare professionals' mental well-being. Post-traumatic stress disorder and depression were found to significantly diminish their peace of mind (PoM), a key factor of emotional resilience, and have wider effects on patient care. The results draw attention to a need for greater mental health support within the Iraqi healthcare system. Peer support, greater accessible psychological services and trauma-informed training may promote well-being for the professional and personal pressures that healthcare professionals carry.

## Introduction

Iraq has endured decades of war, political instability and widespread violence, all of which have severely disrupted the health system and contributed to an escalating mental health crisis (Parkyn and Wall, 2019; Elhadi et al., 2020). Millions of Iraqis, including internally displaced persons (IDPs), endure daily psychological stressors, yet the country's fragile infrastructure remains unable to meet the growing demand for mental health services (UNHCR, 2025). Recent systematic evidence on Iraqi IDPs further documents high levels of trauma and mental health problems in the aftermath of the 2014 Islamic State of Iraq and Syria invasion (Ahmed et al., 2024). Healthcare professionals, in particular, stand at the frontline of this crisis, facing not only the trauma of their patients but also their own chronic exposure to stress, insecurity and resource shortages (Lafta et al., 2021; Mohammad et al., 2021).

Against this backdrop, conditions such as post-traumatic stress disorder (PTSD) and depression have become increasingly common among Iraqi healthcare workers (Ibrahim et al., 2018; Elnakib et al., 2021). These disorders erode what has been described as "peace of mind" (PoM) – a construct that goes beyond the absence of illness to reflect inner calm, emotional balance and psychological stability (Lee et al., 2013). Clarifying this construct is essential, as PoM represents a culturally relevant and context-specific dimension of well-being often disrupted in conflict settings.



Despite the evident burden of psychological distress, Iraq's mental health system remains severely underdeveloped. It suffers from a lack of adequately trained professionals, inadequate integration of mental health into primary care and inadequate institutional policies regarding the well-being of workers (Sadik et al., 2011). These systemic failures leave workers themselves without sufficient support, even as they intensify the pressures they face both professionally and personally. While the bulk of past research has been on the wider population or on refugees, many fewer have explored the mental health of Iraqi workers themselves – a missing piece in the literature. Evidence elsewhere that conflict-terrorized regions like Gaza produce high levels of psychological distress among both workers and civilians (Marie et al., 2020; Alah, 2024; Nyhus et al., 2025) gives an idea of why the situation here must be explored.

Healthcare workers in Iraq operate within a chronically strained health system shaped by conflict, resource shortages and structural stressors (Sadik et al., 2011). Recent assessments highlight persistent gaps in mental health and psychosocial support across the country (Ahmed, 2022). Although international research has documented elevated risks of PTSD and depression among healthcare professionals in crisis-affected settings (Ibrahim et al., 2018; Elnakib et al., 2021), evidence from Iraq remains limited and fragmented. Most studies focus on the general population or post-conflict community trauma rather than on the psychosocial burden borne by healthcare workers themselves, particularly in relation to workplace demands and emotional strain (Lafta et al., 2021; Mohammad et al., 2021). This study, therefore, addresses this gap by examining PTSD, depression and PoM among Iraqi healthcare workers and by exploring how these outcomes relate to key demographic and workplace characteristics.

Furthermore, current studies often lack theoretical integration. By applying the Conservation of Resources (COR) theory (Hobfoll et al., 2007) and the Job Demands–Resources (JD-R) model (Bakker and Demerouti, 2007), this study offers a more comprehensive framework for understanding how scarce resources, high demands and chronic exposure to violence shape psychological outcomes. This theoretical lens provides a nuanced explanation of how PTSD and depression undermine PoM, while also identifying potential protective factors such as age, gender, years of experience and work setting.

In summary, the existing literature points to several critical gaps that justify the present study. First, there is a lack of focused empirical research examining the mental health of Iraqi healthcare workers. Second, PoM has rarely been examined as a distinct psychological construct, rather than a general indicator of well-being. Third, theoretical frameworks have been underutilized in explaining psychological outcomes among healthcare workers operating in conflict-affected environments. Fourth, little is known about how demographic and professional characteristics shape vulnerability to PTSD and depression in this population. Addressing these gaps within the Iraqi context enables this study to contribute evidence that is both academically relevant and practically valuable for policy development and mental health interventions.

**RQ1:** *What are the measured levels of PTSD symptoms, depressive symptoms and peace of mind among healthcare workers employed in hospitals and outpatient clinics across Iraq?*
**RQ2:** *How do PTSD, depressive symptoms and peace of mind differ across key demographic characteristics (age, gender, marital status and educational level) and workplace factors (years of experience, type of facility and work setting)?*

**RQ3:** *To what extent do PTSD symptoms and depressive symptoms statistically predict peace of mind among Iraqi healthcare workers?*

**Based on prior evidence, we hypothesize that:**

**H1:** *Higher levels of PTSD symptoms will be significantly associated with lower levels of peace of mind among Iraqi healthcare workers.*
**H2:** *Higher levels of depressive symptoms will be significantly associated with lower levels of peace of mind among Iraqi healthcare workers.*
**H3:** *Key demographic characteristics (age, gender, marital status and educational level) and workplace factors (years of experience, type of facility and work setting) will show significant differences in PTSD symptoms, depressive symptoms and peace of mind.*
**H4 :** *PTSD symptoms and depressive symptoms will significantly predict peace of mind in multivariable regression models.*

### Theoretical framework

Healthcare workers in conflict settings bear a dual burden: treating victims of war-related trauma while enduring their own psychological distress. Evidence shows that repeated exposure to violence, high patient loads and ethical dilemmas elevate risks of PTSD, depression and anxiety (Miller and Rasmussen, 2010; Ager et al., 2012; Elnakib et al., 2021). In Iraq, these risks are intensified by decades of instability, shortages of resources and the neglect of mental health services (Sadik et al., 2011; Parkyn and Wall, 2019).

PTSD is the most common condition in such situations. Intrusive memories and hyperarousal symptoms interfere with functioning and well-being (Charlson et al., 2012). Iraqi research substantiates high rates among clinicians, mainly propelled by repeated trauma, as well as the absence of institutional support (Ibrahim et al., 2018; Bou-Orm et al., 2023). Having fragile infrastructure bases and limited trained health workers, most providers are continuously left exposed without the proper care (Elhadi et al., 2020).

Depression is also prevalent as a sequela, most likely related to insecurity, work compressedness and exposure to death. It is accompanied by fatigue, low mood and unprofessional working practice (Mento et al., 2020; Seiler et al., 2024). In Iraq, where violence cases and deteriorating health infrastructure still occur, practicing health professionals deal with high depression symptoms and limited care access (Ibrahim et al., 2018; d'Ettorre et al., 2020).

PoM offers a useful construct for understanding these dynamics. Unlike general well-being, PoM emphasizes inner calm and psychological stability (Lee et al., 2013). In Iraq's conflict environment, PoM requires both individual coping strategies and institutional backing (Van Ommeren et al., 2015; Bou-Karroum et al., 2020). Yet scarce resources and repeated trauma undermine this balance, leaving providers vulnerable to burnout and emotional strain.

Recent Gaza research also established that health professionals who suffered chronic conflict stress and war stressors experienced high rates of PTSD, depression and lost resilience (Alah, 2024; Nyhus et al., 2025). These results provide support to the value of the "PoM" as a valuable construct, as the recurrent trauma and restricted institutional resources erode both personal coping and professional functioning. By connecting the results to the Iraqi case, the current research situates the conceptual foundations within broader regional research, increasing the theories' and the COR's, as well as the JD-R, transferability (Marie et al., 2020).

To describe these patterns, this research uses the COR theory and the JD-R framework. COR theory suggests that stress occurs where valued resources, like support or job stability, are threatened or lost (Hobfoll et al., 2007). The JD-R framework emphasizes the ways that high demands and limited resources combine to forecast outcomes such as burnout and decreased well-being (Bakker & Demerouti, 2007). Taken together, these theories help us to see why Iraqi health workers, experiencing high demands and limited resources, are at high risk for PTSD and depression, and why their PoM is brittle.

Overall, the literature reveals consistent vulnerabilities but leaves gaps regarding how PTSD and depression jointly affect PoM, and whether demographic and occupational factors moderate these effects. This study addresses that gap within Iraq's unique conflict context.

## Methods

### Study design and setting

This study adopted a quantitative cross-sectional design to evaluate psychological well-being among Iraqi healthcare professionals. Data were collected over a 2-month period in 2024 via an online survey (Google Forms) to maximize reach and participation. The sampling frame included health workers across public and private hospitals and outpatient clinics in central and southern governorates, with limited representation from the Kurdistan Region of Iraq (KRI). This ensured coverage of a range of institutional settings, including hospitals and outpatient facilities, thereby capturing perspectives from different levels of healthcare delivery. The design and reporting of study components – setting, participants, variables, measurement and statistical analysis – were guided by the Strengthening the Reporting of Observational Studies in Epidemiology Statement (Von Elm et al., 2007) to ensure clarity, transparency and reproducibility.

### Participants

This study recruited healthcare professionals from multiple regions of Iraq, with the majority of responses originating from the central and southern governorates and limited participation from the KRI. Eligible participants comprised physicians, nurses, pharmacists and allied health workers employed in public or private hospitals or outpatient clinics. Eligibility criteria were: age $\geq$ 18 years; active practice in Iraq during the data-collection period; and provision of informed consent. Exclusion criteria were: not being directly involved in healthcare delivery and incomplete submissions that did not meet a predefined completeness threshold.

The geographical distribution of participants was based on self-reported employment locations. A total of 133 healthcare workers were employed in facilities located in the central governorates and 41 in the southern governorates. These figures reflect the primary areas from which respondents were obtained and form the basis for the regional classification used in the analysis. Thirteen respondents indicated personal origin or affiliation with the KRI; however, all of them reported that their current workplace was situated within the central governorates. Because this information was derived from participants' own responses, and the KRI-related subsample was too small to support a separate regional analysis, these cases were analytically retained within the central governorates category to avoid misclassification and overinterpretation. Within the central governorates, a small number of participants

reported personal ties to KRI; however, because all were employed in central-region facilities, they were analytically retained within this category to avoid unstable subgroup estimates.

The final analytic sample comprised 174 participants spanning diverse professional roles, years of experience, genders and institutional affiliations. This heterogeneity enriches the descriptive scope of the study and provides valuable insights across varied institutional and regional contexts, although generalizability remains limited by the sampling frame and the under-representation of KRI.

### Sample size justification

A precision-based calculation for a single proportion at a 95% confidence level with a conservative $p = 0.50$ and margin of error $d = 0.075$ yielded $n = Z^2 \cdot p(1 - p)/d^2 \approx 171$ (Cochran, 1977). The achieved sample of $N = 174$, therefore, met this requirement. Before data collection, this target served as the minimum threshold to ensure acceptable precision in estimating key psychosocial outcomes and to strengthen the robustness of the analytical models. For the planned multiple linear regression with up to eight candidate predictors, the achieved $N$ also satisfied commonly cited adequacy rules (e.g., $N \geq 50 + 8 m$ for testing the overall multiple correlation; $N \geq 104 + m$ for testing individual predictors), supporting model stability and precision (Green, 1991; ANKERST, 2016). This explicit justification is consistent with reporting guidance for observational studies that encourages authors to state how sample size was determined (Von Elm et al., 2007).

### Survey instrument

A structured questionnaire was developed for this study, consisting of demographic questions and standardized psychological assessment scales. The questionnaire was initially created in English and subsequently translated into Arabic to ensure clarity and comprehension among participants. The translation process followed rigorous back-translation methods to maintain the accuracy of the survey items. The survey included the following components:

1. **Demographic information:** Participants provided information on age, gender, marital status, educational level, geographic location, workplace setting, specialty and years of experience. Age was categorized into three groups: 18–24, 25–34 and $\geq$ 35 years. Marital status was initially collected in broader categories, but due to the limited representation of divorced, widowed or separated individuals, these responses were collapsed into two analytic groups: married and single. Educational levels included both high school graduates and bachelor's degree holders, reflecting the Iraqi healthcare system, where certain healthcare professionals, such as technicians and assistants, may not hold a university degree but are actively engaged in clinical practice. In Iraq, a proportion of health workers – including medical assistants, midwives and health technicians – enter the workforce after completing secondary education and accredited vocational training programs. Therefore, the 7% of participants with only a high school education accurately represent this segment of the national healthcare workforce. Geographic coverage was classified into three regions: central governorates, southern governorates and the KRI. These demographic variables were considered relevant because prior research indicated that age, marital status, education and geographic location are associated with differences in psychological

well-being, coping capacity and resilience among healthcare workers.

2. **Psychological assessments:**

*PTSD Scale (Diagnostic and Statistical Manual of Mental Disorders, Fourth Edition [DSM-IV]):* Symptoms of PTSD were assessed using the PTSD Scale based on DSM-IV diagnostic criteria (American Psychiatric Association, 1994). The instrument has been widely used in clinical and research settings to screen for PTSD symptoms, demonstrating strong construct validity and reliability across diverse populations (Foa et al., 1993; Weathers et al., 2001). In the current study, the scale demonstrated high internal consistency, with a Cronbach's alpha of 0.91.

*Beck Depression Inventory (BDI):* Depressive symptoms were rated on the BDI (Beck et al., 1961), the most common measure utilized to rate the degree of depression. We made use of the proven Arabic translations that were provided by Abdel-Khalek (1998) and by Al-Musawi (2001), who both established the level of their degree of correspondence and construct validity among Arabic speakers. Internal consistency was very high within the present sample on the BDI, with its Cronbach's alpha being 0.89. Clarification: The term utilized as "Depression" throughout the Results section and the Discussion refers specifically to the BDI scores received. As such, the terms and phrases of both terms run freely throughout the manuscript.

*PoM Scale:* The PoM Scale, originally developed by Lee et al. (2013), was used to assess participants' inner peace and emotional balance. The scale has shown strong psychometric properties across cultures, including satisfactory validity and reliability indices (Lee et al., 2013). In this study, the PoM Scale yielded a Cronbach's alpha of 0.87, indicating good internal consistency.

### Data collection

Data were collected over a 2-month period. To broaden reach and enhance sample diversity, the online survey (Google Forms) was disseminated via official mailing lists of national healthcare associations, hospital administrative distributions and professional social media platforms commonly accessed by healthcare workers. A nonprobability convenience sampling strategy, supplemented by snowball recruitment, was therefore employed to reach healthcare workers across different institutions and regions. Respondents were additionally invited to share the survey link with colleagues (snowball sampling). In the absence of a central national registry of healthcare professionals, a nonprobability convenience sampling approach was employed. While this method enabled efficient access to a wide spectrum of roles, specialties and settings, it entails potential selection and nonresponse biases; therefore, the findings should not be interpreted as nationally representative. The online survey link was shared through email to healthcare personnel from public and private hospitals. The participants' involvement in the study was voluntary, after having provided informed consent online. To increase responses, reminder emails were sent every 2 weeks. Participants accessed the survey through a secure online link and completed the questionnaire anonymously. No personally identifiable information was collected, and all responses were stored on a password-protected database accessible only to the research team. Completion time for the survey was ~10–15 min.

### Data analysis

Data analysis was conducted with the utilization of Statistical Package for the Social Sciences version 25.0. Demographic characteristics were described in the methods section. The mean scores and the standard deviations (SDs) were employed to evaluate the three main outcomes of PTSD, depression and PoM. To test the effects of demographic variables on the psychological outcomes, inferential statistics such as $t$-tests and Spearman's rank correlation coefficients were used. Furthermore, a step-wise linear regression was performed to establish how PoM is predicted among healthcare workers. In line with best practices for reporting scale-based outcomes, the BDI was analyzed using established clinical cutoff scores to categorize severity (0–13 minimal, 14–19 mild, 20–28 moderate and ≥29 severe), consistent with the original instrument and its Arabic validation studies. For the PTSD (DSM-IV, 17 items), population-specific diagnostic thresholds vary across contexts; therefore, descriptive severity bands (minimal/mild/moderate/ severe) were applied to summarize symptom burden without implying clinical diagnosis, and raw means and SDs were also reported. As the PoM scale has no validated clinical cutoffs, low, moderate and high categories were derived from sample-based distributional thresholds to aid interpretability, alongside mean and SD values. To contextualize the observed central tendencies, one-sample $t$-tests were conducted against theoretical midpoints (PTSD = 34; BDI = 31.5; PoM = 21). In this study, the hypothesized means (34 for PTSD, 31.5 for depression and 21 for PoM) were derived from prior validation studies and normative values reported in the literature for comparable populations (Foa et al., 1993; Beck et al., 1961; Lee et al., 2013). These values served as reference benchmarks to determine whether the mean scores of Iraqi healthcare workers significantly deviated from established standards.

## Results

### Sample characteristics

Table 1 summarized the characteristics of 174 healthcare professionals in Iraq, encompassing a range of age groups, marital status, educational backgrounds, workplace settings and years of experience.

A majority of the participants were male (64.4%, $n = 112$), with females representing 35.6% of the sample ($n = 62$). In terms of age distribution, the largest group was those aged 35 years and older (57.5%, $n = 100$), followed by those aged 30–34 (25.9%, $n = 45$), 25–29 (12.6%, $n = 22$) and 20–24 years (4.0%, $n = 7$). Most of the participants were married (79.3%, $n = 138$), with single participants comprising 20.7% of the sample ($n = 36$).

With respect to educational levels, 51.7% held a bachelor's degree ($n = 90$), 20.7% had a master's degree ($n = 36$), 20.1% had a PhD ($n = 35$) and a small portion had completed high school (7.5%, $n = 13$). Taking into consideration the geographical distribution, 76.4% of the healthcare workers were from the Central Governorates ($n = 133$), while 23.6% were from the Southern Governorates ($n = 41$).

Regarding employment setting, a substantial number of participants worked in hospitals (76.4%, $n = 133$), while 23.6% were employed in clinics ($n = 41$). In terms of specialty, 36.2% of the participants were in General medicine ($n = 63$), 25.3% in Nursing ($n = 44$), 22.4% in Mental health ($n = 39$) and 16.1% in Surgical and emergency ($n = 28$).

Experience levels among the participants also varied, with 38.5% having more than 15 years of experience ($n = 67$), 25.3% having 10–15 years ($n = 44$), 16.7% having 5–10 years ($n = 29$) and 19.5% having 1–5 years of experience ($n = 34$).

**Table 1.** Demographic characteristics of healthcare workers in Iraq (N = 174)

| Characteristic | | N | % |
|---|---|---|---|
| Gender | Female | 62 | 35.6 |
| | Male | 112 | 64.4 |
| Age (years) | 20–24 | 7 | 4.0 |
| | 25–29 | 22 | 12.6 |
| | 30–34 | 45 | 25.9 |
| | 35+ | 100 | 57.5 |
| Marital status | Single | 36 | 20.7 |
| | Married | 138 | 79.3 |
| Education | High school | 13 | 7.5 |
| | Bachelor | 90 | 51.7 |
| | Master | 36 | 20.7 |
| | PhD | 35 | 20.1 |
| Location | Southern governorates | 41 | 23.6 |
| | Central governorates | 133 | 76.4 |
| Facility | Clinic | 41 | 23.6 |
| | Hospital | 133 | 76.4 |
| Specialty | Mental health | 39 | 22.4 |
| | General medicine | 63 | 36.2 |
| | Nursing | 44 | 25.3 |
| | Surgical and emergency | 28 | 16.1 |
| Experience | 1–5 Years | 34 | 19.5 |
| | 5–10 Years | 29 | 16.7 |
| | 10–15 Years | 44 | 25.3 |
| | >15 Years | 67 | 38.5 |

According to Table 2, the results from the PTSD (DSM-IV) scale indicated a range of symptom severity among the participants: 32.8% (n = 57) exhibited no to minimal PTSD symptoms, the largest portion, 50.6% (n = 88), experienced mild PTSD, 15.5% (n = 27) had moderate PTSD and only 1.1% (n = 2) suffered from severe PTSD.

### Descriptive outcomes and severity classifications

Descriptive statistics and severity distributions for PTSD, depression (based on established BDI cutoffs) and PoM (distribution-based categories) are shown in Table 2; one-sample $t$-tests assessed deviations from theoretical midpoints.

For depression, assessed via BDI, most participants showed minimal depressive symptoms, with 60.9% (n = 106) falling into this category. Mild depression was reported by 20.1% (n = 35), moderate depression by 13.8% (n = 24) and severe depression by 5.2% (n = 9). These categories reflect the established BDI clinical cutoffs (0–13 minimal, 14–19 mild, 20–28 moderate and ≥29 severe). In terms of PoM, 10.3% (n = 18) of the sample reported low PoM, whereas 54.0% (n = 94) achieved a moderate level and 35.6% (n = 62) reported high PoM. Because no validated clinical cutoffs exist for PoM, these categories were derived from sample-based distributional thresholds. For PTSD, the mean score was 22.29 (SD = 12.19), and participants were described across minimal, mild, moderate and severe symptom bands. These PTSD categories are descriptive only and do not imply diagnostic status, as cutoff thresholds vary across populations.

A one-sample $t$-test was conducted to determine whether the mean scores of PTSD, depression and PoM were significantly different from the hypothesized means. For PTSD symptoms, the mean score was 22.29 (SD = 12.19) with a hypothesized mean of 34, resulting in a $t$-test value of −12.67, indicating that the mean PTSD score was below the hypothesized mean. The mean depression score was 11.64 (SD = 9.20) with a hypothesized mean of 31.5, resulting in a $t$-test value of −28.48, indicating that the mean depression score was also below the hypothesized mean. Conversely, the mean score for PoM was 23.48 (SD = 5.99) with a hypothesized mean of 21, resulting in a $t$-test value of 5.47, indicating that the mean PoM score was higher than the hypothesized mean.

The study explored the relationships between several demographic variables and psychological outcomes using Spearman's rank correlation coefficient. The results indicate significant correlations, highlighting how age and years of experience influence

**Table 2.** Descriptive statistics and one-sample t-test results for PTSD, depression and peace of mind among healthcare workers in Iraq (N = 174)

| Scales | Items | Level | N | % | Mean | SD | Hypothesized mean | t (173) |
|---|---|---|---|---|---|---|---|---|
| PTSD | 17 | Minimal | 57 | 32.8 | 22.29 | 12.19 | 34 | −12.67*** |
| | | Mild | 88 | 50.6 | | | | |
| | | Moderate | 27 | 15.5 | | | | |
| | | Severe | 2 | 1.1 | | | | |
| Beck's depression inventory | 21 | Minimal | 106 | 60.9 | 11.64 | 9.20 | 31.5 | −28.48*** |
| | | Mild | 35 | 20.1 | | | | |
| | | Moderate | 24 | 13.8 | | | | |
| | | Severe | 9 | 5.2 | | | | |
| Peace of mind | 7 | Low | 18 | 10.3 | 23.48 | 5.99 | 21 | 5.47*** |
| | | Moderate | 94 | 54 | | | | |
| | | High | 62 | 35.6 | | | | |

Note: ***$p$ < 0.001.

PTSD symptoms and PoM, whereas educational levels appear less correlated with these psychological outcomes.

### Correlations

Bivariate associations between demographic variables and psychological outcomes were examined using Spearman's rank correlations (Table 3).

The PTSD scale (DSM-IV) exhibited a significant negative correlation with the age group ($r = -0.197$, $p < 0.05$) and years of experience ($r = -0.176$, $p < 0.05$). This suggested that older healthcare workers and those with more years of experience tend to report fewer symptoms of PTSD. Conversely, there was no significant correlation observed between PTSD symptoms and education level ($r = 0.031$), indicating that educational attainment may not significantly influence the prevalence or intensity of PTSD symptoms among the healthcare workers in this context.

BDI scores were not significantly correlated with age group ($r = -0.128$), education level ($r = 0.041$) or years of experience ($r = -0.146$). These findings suggest that depression among healthcare workers in conflict zones may be influenced by factors other than the demographics considered in this study.

In contrast, PoM showed a pattern of significant correlations. It was positively correlated with age group ($r = 0.169$, $p < 0.05$) and years of experience ($r = 0.214$, $p < 0.01$), indicating that older workers and those with more experience tend to achieve higher levels of PoM. Interestingly, there was a significant negative correlation between PoM and education level ($r = -0.164$, $p < 0.05$), suggesting that higher educational levels might be associated with lower PoM.

Independent-samples *t*-tests were conducted (Table 4) to evaluate the differences in psychological measures (PTSD, depression and PoM) between healthcare workers in two types of facilities: hospitals and clinics.

### Group differences by facility

Independent-samples *t*-tests compared outcomes by facility type (hospital vs. clinic) (Table 4).

The mean PTSD score was significantly higher in hospitals ($M = 23.35$, $SD = 12.05$) compared to clinics ($M = 18.83$, $SD = 12.17$), $t(172) = 2.10$, $p < 0.05$. This suggested that healthcare workers in hospital settings may experience higher levels of PTSD symptoms than their counterparts in clinics. For depression, the results indicated a higher mean score in hospital settings ($M = 12.26$, $SD = 9.47$) compared to clinics ($M = 9.61$, $SD = 8.03$), though this difference was not statistically significant, $t(172) = 1.62$, $p > 0.05$. Conversely, PoM scores were found to be lower in hospitals ($M = 23.04$, $SD = 5.86$) than in clinics ($M = 24.93$, $SD = 6.25$). The negative *t*-value, $t(172) = -1.78$, $p < 0.05$, indicated a significant difference, with clinic workers reporting higher PoM than those in hospitals.

**Table 3.** Spearman correlation coefficients between demographic characteristics and psychological scales among healthcare workers

| Spearman correlation | Age group | Education level | Year of experience |
|---|---|---|---|
| PTSD Scale (DSM-IV) | −0.197** | 0.031 | −0.176* |
| Beck's Depression Inventory | −0.128 | 0.041 | −0.146 |
| Peace of Mind | 0.169* | −0.164* | 0.214** |

Note: *p < 0.05, **p < 0.01.

**Table 4.** Independent samples *t*-test results for psychological measures by facility type

| Scales | Type of facility | Mean | SD | t (172) |
|---|---|---|---|---|
| PTSD | Hospital | 23.35 | 12.05 | 2.10* |
| | Clinic | 18.83 | 12.17 | |
| Beck | Hospital | 12.26 | 9.47 | 1.62 |
| | Clinic | 9.61 | 8.03 | |
| Peace | Hospital | 23.04 | 5.86 | −1.78 |
| | Clinic | 24.93 | 6.25 | |

Note: *p < 0.05.

Furthermore, additional analyses considering other demographic variables, such as city (location), gender, marital status and speciality, revealed no significant differences, and results are detailed in Appendix A.

### Regression analysis

Table 5 summarized a stepwise linear regression fitted to identify independent predictors of PoM) among healthcare professionals. Candidate predictors included PTSD, depression (BDI), age, years of experience, education level, marital status, facility type (hospital vs. clinic) and region (central, southern and KRI). The final model retained only variables with statistically significant coefficients; retained estimates and model-fit statistics are reported in Table 5.

The regression analysis showed significant predictors of PoM among healthcare workers. The adjusted $R^2$ of 0.518 ($F(4, 172) = 47.536$, $p < 0.001$) indicated that ~52% of the variance in PoM can be explained by the model. Depression scores from BDI emerged as a particularly strong predictor, with higher scores significantly associated with lower PoM ($B = -0.369$, $p < .001$). Additionally, PTSD symptoms negatively influenced PoM ($B = -0.082$, $p = 0.008$). Education and experience also played critical roles; more years of education were linked to lower PoM ($B = -1.145$, $p = 0.002$), while more years of experience contributed positively ($B = 0.786$, $p = 0.007$).

### Discussion

From a practical perspective, this study highlights the urgent need for targeted psychological interventions to support healthcare workers in conflict zones. The results indicate the dual influence of psychological stressors and professional resources on PoM, underscoring the importance of institutional support systems and individual resilience-building strategies. These insights provide actionable directions for organizational policies and mental health programs designed to mitigate PTSD and depression while

**Table 5.** Summary of the final regression model predicting peace of mind

| Predictor | B | SE | t-value | p-value |
|---|---|---|---|---|
| PTSD | −0.082 | 0.031 | −2.675 | 0.008 |
| Beck | −0.369 | 0.041 | −9.101 | <0.001 |
| Education level | −1.145 | 0.366 | −3.131 | 0.002 |
| Years of experience | 0.786 | 0.29 | 2.712 | 0.007 |

Note: $R^2 = 0.529$, Adjusted $R^2 = 0.518$, $F(4, 172) = 47.536$, $p < 0.001$.

enhancing PoM among healthcare professionals working in high-stress environments.

The study demonstrates that healthcare providers in Iraq face a considerable psychological burden, with varying degrees of PTSD and depression negatively affecting overall mental well-being. The presence of mild to moderate symptoms reflects persistent stressful conditions. Conversely, there were individuals with some demonstrating moderate to extreme PoM that deflected access to resources with coping mechanisms to some. These findings indicate the importance of understanding insight to specific psychiatric issues and, more significantly, the supply aid to health care workers affected by war. Institutional remedies, including proper resources, protection and psychiatric care, should comprise the first line, irrespective of individual coping, with that of colleagues supporting.

PoM emerged as a central construct in this study, reflecting participants' ability to maintain inner calm and psychological stability despite ongoing exposure to conflict. The finding that some healthcare workers reported moderate to high levels of PoM, even alongside PTSD and depression symptoms, highlights the role of individual coping strategies, cultural resilience and spiritual frameworks that may buffer against total psychological collapse. This aligns with Lee et al. (2013), who conceptualized PoM as distinct from general well-being, emphasizing balance and inner harmony. In conflict-affected contexts, such as Iraq, sustaining PoM may represent a protective factor that mitigates the impact of resource depletion and overwhelming job demands, offering practical insight for interventions aimed at resilience-building strategies.

### Summary and comparison with existing literature

Rates of PTSD as recorded here are similar to those found in other studies that presented high rates of psychological distress among health workers (Pappa et al., 2020; Shechter et al., 2020). It extends the literature a little further by attributing the contribution of armed conflict, violence and so forth, to the mental health of health workers.

Likewise, previous studies also presented high psychological distress in public health crises, including depression, anxiety, and PTSDs (Kisely et al., 2020; Lai et al., 2020). The current results support these findings and underscore the excess burden of violence related to wars, as participants who were exposed to conflict indicated high rates of PTSD. This is similar to the findings of Alhasnawi et al. (2009), who indicated that exposure to violence was significantly related to the beginning of mental disorders, as well as PTSD in particular.

Qualitative estimations revealed that the health workers were frustrated to be experiencing pressures simultaneously from the pandemic and armed conflict, making their psychological trauma even worse. This is also the same trend as the previous research depicting the aggregated effect among double-strained frontline workers (Chen et al., 2020; Greenberg et al., 2020).

### Application of theoretical models

These findings can be interpreted through the lens of COR Theory, which posits that the loss of resources leads to heightened stress (Hobfoll and Freedy, 2017). The evidence suggests that emotional and psychological resource depletion due to conflict and occupational strain contributed to burnout and PTSD, consistent with prior studies (Rija et al., 2022).

Even the JD-R model offers a helpful framework. It points up that job demands (the high pressure of work, emotional pressure, etc.) will lead to burnout, as well as to the problems of health, if resources (supporting social connections, proper training, etc.) are deficient (Bakker & Demerouti, 2007). This study corroborates the JD-R model, as primary healthcare workers in Iraq encountered overwhelming job demands with insufficient resources, which was reflected in high levels of PTSD.

### Implications for practice and policy

These findings have important implications for practice and policy across healthcare, social work and psychology, particularly in conflict-affected settings. Healthcare workers who are repeatedly exposed to war-related violence face complex and cumulative stressors that require integrative responses extending beyond individual-level clinical care. Accordingly, effective practice must connect mental health interventions with psychosocial support mechanisms, coordinated referral pathways and organizational strategies that promote staff well-being and sustainability within healthcare institutions (Allen and Spitzer, 2015; Kemp et al., 2015; Nicholas et al., 2019).

Within this context, social work has a distinct and actionable role in supporting Iraqi healthcare workers exposed to chronic conflict-related stressors and psychological distress. In healthcare settings, social workers contribute through psychosocial assessment, care coordination and organizational advocacy, positioning them to identify risk, facilitate confidential referrals and provide brief supportive interventions within multidisciplinary teams (Allen and Spitzer, 2015; Nicholas et al., 2019). In environments marked by ongoing violence and insecurity, repeated exposure to traumatic events can increase vulnerability to PTSD and depression while simultaneously depleting coping resources and social supports, underscoring the need for structured, workplace-based psychosocial responses (Savitsky et al., 2009; Miller and Rasmussen, 2010). Drawing on COR theory and the JD-R model, social work-led initiatives can be conceptualized as resource-restoration strategies that buffer high job demands, support recovery following traumatic exposure and strengthen protective factors at both individual and organizational levels (Hobfoll et al., 2007; Bakker & Demerouti, 2007). This approach is consistent with guidance on mental health and psychosocial support in emergency and conflict settings, which emphasizes strengthening functional and social resources alongside distress reduction, and supports positioning PoM as a meaningful psychosocial outcome targeted through culturally responsive institutional supports (Baingana et al., 2005; Sadik et al., 2011; Lee et al., 2013; Van Ommeren et al., 2015).

From a policy perspective, prioritizing the mental health and psychosocial well-being of healthcare workers is essential for sustaining health systems in protracted crisis contexts. Policies that integrate trauma-informed care principles, workforce training and institutional support structures can mitigate burnout and secondary traumatic stress, conditions commonly observed among healthcare professionals and social workers operating in high-demand environments. Embedding psychosocial care as a routine component of healthcare practice, rather than as an ad hoc response, would reinforce the recognition that mental health is as fundamental as physical health. Such an approach can enhance coping capacity, professional resilience and the quality of care provided to populations affected by war and displacement.

## Strengths and limitations

A significant strength of this study was its cross-sectional quantitative design, which generated reliable evidence on the prevalence and correlates of PTSD among healthcare workers. This approach provided a broad understanding of the problem and allowed for a deeper investigation of the context within the constraints of quantitative analysis. Another strength lies in the use of validated and culturally adapted instruments (PTSD Scale, BDI and PoM Scale), all of which demonstrated strong internal consistency in the current sample. Additionally, the integration of COR theory and the JD-R model as guiding frameworks enhanced the theoretical depth and interpretability of the findings.

However, it is difficult to assume causality due to the cross-sectional design. Furthermore, the sample size was appropriate for the aims of this study, but it may restrict the extent of applicability of the results. Specifically, surgical and emergency healthcare workers were underrepresented in the sample. While their limited numbers restrict generalizability, the findings remain valuable for understanding overall psychological trends across healthcare personnel in conflict settings. Future research should purposively target these frontline groups, as they are likely to experience higher levels of PTSD and depression given their direct and intensive exposure to war-related injuries and emergencies.

A further limitation relates to regional representation. Although the study aimed to include healthcare workers from across Iraq, participation from facilities located directly within the KRI was limited. Thirteen respondents reported personal affiliation with KRI; however, all were employed in institutions situated in the central governorates. As a result, the study cannot offer region-specific estimates for KRI, and the findings should not be interpreted as nationally representative across all Iraqi regions. Nevertheless, including these respondents broadened the geographic diversity of the overall sample and contributed to a more comprehensive national perspective.

Another limitation concerns the reliance on self-reported measures, which may introduce reporting bias or underestimation of symptoms due to stigma surrounding mental health. Despite these challenges, anonymity and confidentiality measures were implemented to reduce bias and encourage honest responses.

In terms of implications, the strengths of this study – its validated tools, theoretical integration and focus on an under-researched population – provide a foundation for developing targeted interventions and informing policy. However, the identified limitations highlight the need for future longitudinal and mixed-methods studies to establish causal relationships, capture deeper contextual insights and ensure broader representation of high-risk subgroups.

## Conclusion

This study demonstrates that Iraqi healthcare workers face significant psychological burdens, with PTSD and depression negatively influencing PoM, while professional experience appears protective. Integrating the COR Theory and the JD-R Model provided a nuanced understanding of how demands and limited resources shape outcomes. These findings underscore the urgent need for institutional mental health support and resilience-building strategies. Policymakers and healthcare organizations should prioritize interventions that sustain workers' psychological well-being. Future longitudinal studies are essential to capture the long-term effects of conflict-related stressors on healthcare professionals.

**Open peer review.** To view the open peer review materials for this article, please visit http://doi.org/10.1017/gmh.2026.10136.

**Data availability statement.** The datasets and codes that support the findings of this study are available from the corresponding author upon reasonable request.

**Acknowledgments.** This work was supported by Open Access funding from Qatar National Library.

**Author contribution.** All authors contributed to the study since conception, methodology, and analysis.
Data collecting by LA and AF. Literature review by YS and YM, and analysis was done by AF, YS and YM, and revisions and supervision were done by YS and AF. All authors approved the final manuscript.

**Financial support.** No funding was received from any source.

**Competing interests.** The authors declare none.

**Ethical considerations.** The study was conducted in accordance with the ethical principles of the Declaration of Helsinki. Ethical approval was obtained from the institutional review board of Ministry of Health (approval code: SH2024-4-4-4370). All participants provided informed consent before participating in the study. Participation was voluntary, and participants had the right to withdraw at any stage without any consequences. Data confidentiality and anonymity were strictly maintained throughout the study.

**Consent to publish.** Participants were informed that the findings of this research would be published in academic journals and potentially presented at conferences. Consent for publication was obtained from all participants prior to their inclusion in the study.

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
