## [Reviewer Report]

The study aims to evaluate PTSD and depression as predictors of Peace of Mind among healthcare workers working with war victims, in Iraq. Although there are limited studies on the region of interest, however I find the manuscript needs a “Major revision” before being considered for publication. The comments are:

1. The introduction section looks too much over-written, including the sub-sectioning of details? The same has been done in the discussion section as well? Why is this required?

2. The authors circulated an online questionnaire for the study conducted? So, was the questionnaire vetted & approved by the respective experts for the quality?

3. In the abstract, method section, “Beck Depression Inventory” is mentioned. However, throughout the manuscript “Depression” has been discussed? Are the two terms same?

4. In Table-1, the “Surgical and Emergency” responders were the least of the population which participated in the study. However, given a war-like situation, such healthcare responders are highly expected to be extensively involved & subsequently burdened with both PTSD and depression. So, how relevant would be the study findings with respect to this limited sample size of such healthcare workers?

5. The abbreviation “PTSD” needs to be followed throughout the manuscript.

6. The authors should consider discussing the “Limitations, strengths, and implications” of the study in a more comprehensive manner. For instance, the limitation of sample size etc.?

---

## [Reviewer Report]

Nice topic, yet major points need to be addressed

-Please specify more in the inclusion criteria, and add the specific setting

-Clarify how you get the sample number according to what?

-The literature is limited and not recent, and no studies about similar regions like Gaza to support

- Ethical consideration is not addressed

-psychometric properties is not written for the tools

- How 7% of health workers have a high school education!!!

- explain what is the hypothesized mean is on page 13

-discussion lacks the peace of mind point

-explain mixed method on page 19

-suggest revising the conclusion for 5 lines only

Wish you luck

---

## [Reviewer Report]

General Comments on the Introduction

• The introduction is vague, broad, and unfocused. It does not effectively guide the reader toward the research questions and results.

• It is too lengthy and spends too much time on general/global issues rather than the Iraqi context.

• Please revise it to make it concise, coherent, and significant, with a sharper focus on Iraq.

Specifically, the introduction should:

• Emphasize the Iraqi context – including war, violence, instability, prevalence of mental health problems, and challenges faced by IDPs.

• Highlight gaps in mental health services, infrastructure, and human resources and explain how these contribute to the issues you are studying.

• Clearly articulate the research gap and limitations of existing studies.

• State the significance and novelty of your research.

• Present the research questions and hypotheses directly.

• Provide more local references on Iraq (mental health, war, conflict, IDPs, etc.) rather than relying heavily on global studies.

- For example, I noticed that Ibrahim et al., 2018 was repeatedly cited incorrectly – that paper validated a PTSD scale but did not cover what was claimed in your text.

- Please conduct a more comprehensive search for relevant Iraqi studies.

Specific Comments on Sentences in the Introduction:

• “Among the various dimensions of well-being, peace of mind—defined as a state of mental and emotional tranquility—is significantly influenced by conditions such as PTSD and depression...” → This is vague. Please clarify the meaning of “peace of mind” (e.g., do you mean well-being?) and support with appropriate references.

• “Healthcare workers in Iraq face significant stress management challenges...” → This statement is exaggerated. It must be supported with data and references.

• “This study provides a novel contribution by examining the relationship...” → This paragraph is too fancy and not placed correctly. It should be integrated into the rationale, significance, and objectives section. Avoid overstating claims that your study cannot fully address.

Methods Section

• Study design and setting: Very vague and unclear. Please describe in detail and cite a reporting guideline such as STROBE.

• Clarify whether this is a national study or a survey limited to specific areas. What are the eligibility criteria?

• Participants: Expand on recruitment procedures, sampling strategy, and justification.

• Measures: Provide details for each scale (origin, validity, reliability, Cronbach’s alpha in the current study, with references).

• Demographics: Clarify how age groups were classified (e.g., what does “35+” mean?). Why include both high school graduates and BS holders if the focus is health professionals? For marital status, why only married/single – what about divorced, widowed, or separated? Please explain how these demographic variables are relevant to your research questions and analyses.

• Clarify geographic coverage (South, Central, KRI).

Analysis & Results

• For Table 2, clarify if you used cut-off scores (if scales have them) or only reported means and SD. If cut-offs exist, they should be applied.

• Ensure the methods and results are written in past tense.

• Results should be clearly structured, presenting findings systematically.

Discussion & Abstract

• The discussion needs major revision:

o It should interpret the findings in relation to existing literature, particularly in the Iraqi context.

o Avoid repeating results and avoid unsupported or general global statements.

• The abstract should be revised after you restructure the introduction, methods, and results to ensure consistency.

General Notes

• The manuscript currently lacks sufficient references, and some citations are irrelevant. Please include more local Iraqi references and ensure all are accurate.

• Clearly declare if AI tools were used in writing or analysis.

• Overall, the paper requires stronger focus, consistency, and contextualization within Iraq.

---

## [Reviewer Report]

1. Sampling and Representation

- Please indicate the exact number of samples collected per location. You mentioned that the number of participants from the Kurdistan Region of Iraq (KRI) was limited. If the KRI sample size is small, how did you justify assessing mental health problems among health workers in that region?

- In Table 1, you referred to “southern and central governorates,” but there is no mention of KRI. If your aim is to identify regional differences, the data should be divided based on southern and central governorates (Arab provinces) and KRI.

- It is unclear how the 174 samples were selected and from which specific areas they were obtained. This raises concerns about representativeness. The sampling process needs clearer classification, and a sample size calculation should have been conducted prior to data collection.

2. Study Rationale and Research Objectives

- The rationale for conducting this study remains unclear. The authors should explicitly explain why this study was undertaken, the existing gap in the literature, and the intended outcomes or contributions.

- The research questions are currently vague and not sufficiently promising. They should be more specific, focused, and achievable.

- The research questions and hypotheses are not aligned and must be clearly matched for logical consistency.

3. Clarity of Key Points and Objectives

- The four “key points” presented are confusing. It is unclear why they were added or how they differ from the study objectives and research questions. Please clarify their purpose and relevance.

4. Introduction and References

- The Introduction section lacks recent data and relies heavily on reports from NGOs such as IOM and UNHCR, which are not peer-reviewed. Please strengthen this section by including more recent and scholarly sources.

- Consider reviewing and incorporating the following relevant reference if appropriate for your study:

Trauma and Mental Health Problems among Iraqi IDPs following the 2014 ISIS Invasion: A Systematic Review. https://doi.org/10.1080/13623699.2024.2411651

o Please review your current reference list; some citations appear irrelevant. Ensure that all references are directly aligned with your study context and are up to date.

---

## [Reviewer Report]

Dear authors,

Thank you for addressing the comments and strengthening your study. I have only a few minor suggestions to further improve the clarity and organization of your Methods section:

Please ensure all abbreviations (e.g., STROBE) are written in full at first mention before using the shortened form.

Some information appears under the wrong headings (e.g., parts of data collection under participants, and vice versa). Reorganizing these sections will improve clarity.

The data collection procedure would benefit from slightly more detail to improve transparency.

Ensure consistent structure and alignment with standard observational study reporting.

These are minor points, and overall the manuscript is greatly improved.

Well done, and good luck with the final submission.